# Peer review of "First-Principles Study on Hydrogen Storage Performance of Transition Metal-Doped Zeolite Template Carbon"

_crystals, doi:10.3390/cryst9080397_

Round 1

Reviewer 1 Report

Journal Crystals (ISSN 2073-4352)

Manuscript Number: crystals-550827

TITLE: First principles study on hydrogen storage performance of transition metal-doped zeolite template carbon.

Aiming to study the hydrogen adsorption characteristics and mechanism of transition metal-doped zeolite template carbon as a novel porous material are studied by theoretical calculations employing first principle all-electron atomic orbital method based on density functional theory. The topic is very interesting. Therefore, I recommend that the manuscript be published only after addressing the following queries/suggestion below.

1. I encourage the authors to bring some highlights regarding the clustering of these metals on the zeolite template carbon.

2. I encourage the authors to make a tableau that summarizes the physical and chemical properties like binding energy , bond length  between metal and the substrate and the hydrogen stretching bond length

3. I encourage the authors to study both sides doped with transition metal and the hydrogen uptake.

4. I encourage the authors to make a comparison between their results and what is report on the reference Nanotechnology 30.7 (2018): 075404.

DECISION: Not acceptable in the current form. Recommended for publication after addressing  these corrections in line with the journal  of crystals guidelines

Author Response

Response and Revise

Each review items from Reviewer 1 has been accordingly revised as follows:

1. A complementary paragraph concerning the cluster of metal atoms adsorbed on ZTC is added after the end of third paragraph in Section 3 (line 167) as “A cluster of metal atoms are individually added at every adsorption site on ZTC convex surface with the binding distance similar to single atom coordinating on ZTC, which is modeled as the initial molecular structure and relaxed with geometry optimization to investigate the possibility of atomic accumulations for transition metal dopants. The geometrically optimization results as shown in Figure 3 for a representative cluster of 5 Sc atoms doped on ZTC surface center illustrate that the multiple metal atoms decorated on ZTC will dispersively distribute on adjacent adsorption sites, verifying that the doped transition metal atoms can not assemble together due to the high adsorption force deriving from coordinating interaction with carbon rings of ZTC.” accordingly with the updated Figure 3 being inserted below. 

2. In the 1st paragraph of Section 3 (line 125) below which Table 1 is complemented for listing binding energy and coordination length, the according indications are inserted as “The calculation results of binding energy and coordination bond length (measured by the averaged distance away from nearing-carbon atoms) between doped metal atom and ZTC substrate for all kinds of decoration sites (A1, A3, A4, A10) are listed in Table 1.” However, no appreciable bond stretching of Hydrogen molecules has been found in the H2 loading simulations compared to the isolated single H2 molecule (all by geometrical optimization), for which the bond lengths of H2 molecules are not provided here.

3. Even though the binding energies of doped transition metal atoms at every site on ZTC concave side are as high as those at their counterpart site on ZTC convex surface, as our calculation results that have not necessarily presented in this paper, the H2 uptake (loading number) will not significantly increased by doping metal atoms due to the limited space circumvented by ZTC concave framework. Therefore, in this paper, the transition metal modification of H2 storage performance is exclusively focused on introducing the metal dopants at the convex side without considering metal decoration at the concave side of ZTC curve surface.

4. In consistence with the mechanism of H2 adsorption improvement reported by the Nanotechnology 2018 30(7): 075404 (as added reference of [22]), the according analytical discussions are complemented at the end of 2nd paragraph of section 3 (line 150) as Besides, partial positive charges populated on Sc, Ti and V dopants on ZTC imply that multiple H2 molecules will bind to the doped metal atoms through electrostatic interactions with remarkably higher adsorption energies than directly absorbed on ZTC, as verified in the following text and being consistent with recently reported results for carbon nitride (C3N)[22].

Note: All revisions in the revised version of manuscript have been denoted with Blue color.

Reviewer 2 Report

see attached

Author Response

Response and Revise

English expression are thoroughly amended according to Reviewer 2 and denoted with Blue color.

1. (35) the sentence is revised to “There are several basic ways of hydrogen storage--”.

2. (37) “forming chemical bonds” is modified by “chemical bonds formed”.

3. (50) the first sentence is changed to “Zeolites are hydrated crystalline aluminosilicates, which are constructed --- space, can represent various special --- molecular size pore and ---”.

4. (54) “--zeolites persist some--” is amended by “--zeolites have some--”.

5. (56) “commercial use of zeolites does --” is corrected to “zeolites for commercial use at present do --”.

6. (61) “surface area ratio” is rectified as “specific surface area”.

7. (64) “the palpable cation distribution” is changed to “the intrinsic cation distribution”.

8. (74) After “more obviously”, “compared with main-group metal” is added.

9. (75) “Accordingly inspired by this trigger” is revised to “Triggered by this inspiration, the present paper proposes --

10. (76) “doping three representative transition metal atoms” is redressed as “individually doping three representatives of transition metal”.

11. (87) in the “-- which exhibit as a cambered --”, the “exhibit” is altered to “exhibits”.

12. (120) the “-- decorated on it as by doping --” is rectified to be “-- decorated on it as modeled by doping -”

13. (128) “Therefore, it is reasonable----- ” is amended by “It is reasonable and feasible to fulfill improvement of hydrogen adsorption performance by introducing tightly bridging metal atoms which provide higher binding force to hydrogen molecules than ZTC framework”.

14. (139) “d orbitals in partially unoccupied state” is modified to “partially unoccupied d-orbitals”.

15. (140) “-- there is an opportunity for transition metal atoms to form --” is ameliorated to “-- transition metal atoms will interact with the carbon ring of ZTC by forming a stronger force as the coordinate bonding --”.

16. (149) “-- demonstrate the fractional Coulomb attribute of these transition metal atoms as --” is revised to “-- demonstrate the partial Coulomb attribute of these transition metal atoms as coordinating center being attached to ZTC ligands by relatively strong force --”.

17. (162) “With the increase of atomic number, the electron transfer quantity of the doped metal atom declines --” is modified to “The electron transfer quantity of the doped metal atom declines with the increase of its atomic number”.

18. (179) “Referenced” is rectified by “Referencing”.

19. (184) “It will be proved that the proposed mechanism of transition metal decoration i---the above theoretical presumption.” is rectified as by “The calculation results will give a fundamental evidence that the proposed transition metal decoration is a effective routine to ameliorate hydrogen storage performance of ZTC”.

20. (191) “-- 10 kinds of adsorption configurations --” is revised to “-- ten adsorption configurations --”

21. (191) “tended” in “- H2 molecules tended to symmetrically locate -” is substituted by “tend”.

Round 2

Reviewer 1 Report

Manuscript Number: crystals-550827

TITLE: First principles study on hydrogen storage performance of transition metal-doped zeolite template carbon.

Aiming to study the hydrogen adsorption characteristics and mechanism of transition metal-doped zeolite template carbon as a novel porous material are studied by theoretical calculations employing first principle all-electron atomic orbital method based on density functional theory. The topic is very interesting. Therefore, I recommend that the manuscript be published.

Author Response

Revise Indications

English language and style are checked and appropriately modified for the whole manuscript as following list:

1. line 15, “configurations of doped metal atom” in the abstract has been changed into “configurations of the doped metal atom”.

2. line 20, “individually introduced into ZTC model” in the abstract has been changed into “individually introduced into the ZTC model”.

3. line 21, “atoms are are characterized with the adsorption energy” in the abstract has been changed into “atoms are characterized by the adsorption energy”.

4. line 26, “The present investigation provides theoretical basis and prediction” in the abstract has been changed into “The present investigation provides a theoretical basis and predictions”.

5. line 38, “so as to uptake hydrogen” in the first paragraph of the Introduction has been changed into “to uptake hydrogen”.

6. line 43, inserted “such as”, and line 44 “At present” is substituted by “Moreover”.

7. line 48, “the high pressure hydrogen storage” in the first paragraph of the Introduction has been changed into “the high-pressure hydrogen storage”.

8. line 53, “and can represent” is amended to “representing”, and line 56 “is promised” is modified to “can be”

9. line 65, “the intrinsic cation distribution in zeolites make it” in the third paragraph of the Introduction has been changed into “the intrinsic cation distribution in zeolites makes it”.

10. line 67, “light weight” in the third paragraph of the Introduction has been changed into “lightweight”.

11. line 74, “element” is changed by “metal”, and line77 “representatives” is revised to “representative transition metals”, and line 78 inserted “The”.

12. line 88, “connecting in three-dimensional arrangement” in the first paragraph of the Modeling and Calculation Methodology has been changed into “connecting in a three-dimensional arrangement”.

13. line 90, “fifteen hydrogen atoms at edge” in the first paragraph of the Modeling and Calculation Methodology has been changed into “fifteen hydrogen atoms at the edge”.

14. line 109, “from finite basis set” in the second paragraph of the Modeling and Calculation Methodology has been changed into “from the finite basis set”.

15. line 112, “self-consistent field” in the second paragraph of the Modeling and Calculation Methodology has been changed into “the self-consistent field”.

16. line 129, “between doped metal atom and ZTC substrate” in the first paragraph of the Results and Discussion has been changed into “between the doped metal atom and ZTC substrate”.

17. line 146, “as the results shown in Figure 2” in the second paragraph of the Results and Discussion has been changed into “as shown in Figure 2”.

18. line 165, “total energy of modified ZTC system” in the third paragraph of the Results and Discussion has been changed into “total energy of the modified ZTC system”.

19. line 168, “higher structural stability of doping system” in the third paragraph of the Results and Discussion has been changed into “higher structural stability of the doping system”.

20. line 169, “A cluster of metal atoms are individually added” in the fourth paragraph of the Results and Discussion has been changed into “A cluster of metal atoms is individually added”.

21. line 186, “will give a fundamental evidence” in the fifth paragraph of the Results and Discussion has been changed into “will give fundamental evidence”.

22. line 187, “is a effective routine” in the fifth paragraph of the Results and Discussion has been changed into “is an effective routine”.

23. line 213, “the total energy of a isolated” in the seventh paragraph of the Results and Discussion has been changed into “the total energy of an isolated”.

24. line 221, “ZTC show an decreasing trend” in the eighth paragraph of the Results and Discussion has been changed into “ZTC shows a decreasing trend”.

25. line 222, “at A10 as example” in the eighth paragraph of the Results and Discussion has been changed into “at A10 as an example”.

26. line 243, “are basically identical for A3, A4, A10 sites” in the last paragraph of the Results and Discussion has been changed into “is identical for A3, A4, A10 sites”.

Note: The revised parts in text have been indicated by blue color.

Reviewer 2 Report

As I wrote in the previous report, the paper could be published to keep track of another attempt to improve hydrogen storage capacity of nanoporous carbons. From the point of view of scientific content, the paper is correct.

Author Response

(The authors gave the same response as above.)
